# A Review on Superhydrophobic Surface with Anti-Icing Properties in Overhead Transmission Lines

Bo Li [1], Jie Bai [1], Jinhang He [1], Chao Ding [1], Xu Dai [2], Wenjun Ci [3], Tao Zhu [3], Ruijin Liao [2] and Yuan Yuan [3,*]

1   Institute of Electric Power Science of Guizhou Power Grid Co., Ltd., Guiyang 550000, China
2   State Key Laboratory of Power Transmission Equipment & System Security and New Technology, Chongqing University, Chongqing 400044, China
3   School of Materials Science and Engineering, Chongqing University, Chongqing 400030, China
*   Correspondence: yuany@cqu.edu.cn

**Abstract:** The icing on overhead transmission lines is one of the largest threats to the safe operation of electric power systems. Compared with other security accidents in the electric industry, a sudden ice disaster could cause the most serious losses to electric power grids. Among the numerous de-icing and anti-icing techniques for application, direct current ice-melting and mechanical de-icing schemes require power cuts and other restrictive conditions. Superhydrophobic coating technology has been widely focused for good anti-icing properties, low cost and wide application range. However, the special structure of curved transmission lines, complicated service environments, and variated electric performance could significantly limit the application of superhydrophobic anti-icing coatings on overhead transmission lines. In particular, superhydrophobic surfaces can be achieved by combining the rough micro-nano structure and modification agents with low surface energy. Compared with superhydrophobic coatings, superhydrophobic surfaces will not increase the weight of the substrate and have good durability and stability in maintaining the robust structure to repeatedly resist aging, abrasion, corrosion and corona damages, etc. Therefore, this review summarizes the theoretical basis of anti-icing behavior and mechanisms, influencing factors of anti-icing properties, potential techniques of superhydrophobic surfaces on transmission lines, and, finally, presents future development challenges and prospects of superhydrophobic surfaces in the anti-icing protection of overhead transmission lines.

**Keywords:** overhead transmission line; electric power system; anti-icing; superhydrophobic coatings; superhydrophobic surface; techniques

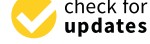



## 1. Introduction

Icing is one of the normal weather conditions possible in daily life on earth. However, icing accretion on surfaces tends to cause overload and damage to equipment operation. Moreover, excessive accumulation of icing and subsequent ice shedding could result in abnormal service conditions, even serious security accidents and economic loss. These disasters often occur in numerous fields [1–5]: aircraft, marine vessels, building constructions, roads of transportation, blades of wind turbines, and power-transmission equipment.

For electric power systems, the safety hazards caused by icing are mainly reflected in two aspects: (i) The flashover power outages caused by the leakage current of iced insulators [6]. (ii) The abnormal motion state of the transmission line and structural damages of the supporting tower caused by the excessive icing accretion on the surface [7]. For icing on insulators, icicles formed during the continuous freezing process could change the potential distribution of the surface. The partial ice melts under the heating effect of the leakage current, resulting in the flashover of insulators [8,9]. For icing on transmission lines, excessive ice accumulation on lines often causes uneven and unbalanced loads on conductors, leading to the conductor galloping, line breakage and tower inversion [10].

Once these accidents happen, severe weather conditions make it difficult to repair the damaged transmission lines, resulting in a long power outage of the power grid and large economic losses [7]. Relevant statistics show thousands of icing accidents on transmission lines since 1950 [11]. Especially in 2008, 12 provincial regions in southern China experienced disastrous snowy weather for more than 20 days in succession, causing serious icing of transmission lines and equipment, eventually causing deterioration to the point of wide-range power system paralysis and economic losses reaching hundreds of billions of yuan.

Therefore, the greatest objective of icing protection for overhead transmission lines is to minimize icing hazards for efficiency and durability and to reduce the technique costs without affecting line operation. Firstly, many scholars have conducted in-depth research on the influencing factors and formation mechanism of ice coverage on transmission lines. Some prevention countermeasures for icing issues have been proposed, such as de-icing techniques [12], optimization of insulator string and conductor structure [13], large current ice-melting equipment [14,15], coating treatment [16,17], etc. Generally, these proposed anti-icing and de-icing techniques on transmission lines all have advantages and disadvantages. De-icing techniques are to realize de-icing effects by external forces, such as de-icing robots. However, the uncontrolled impact of the fallen ice could result in damage to the tower-line system. Some dead corners also have difficulties in operating. In addition, high-strength anti-icing conductors [18] and ice-melting equipment [15] are other schemes for wide adoption in the ice coverage of overhead lines. High-strength icing-resistant conductors show a light weight and high load capacity relative to conventional Al lines, but are restricted in the expensive cost and the transmission level [19]. Moreover, the large current ice-melting equipment is the most applied technique for highly efficient de-icing performance. However, the ice melting technique is mainly applied at 500 kV or above and developing a de-icing method for medium and low-voltage transmission lines [15] is a technical difficulty. In addition, when the ice-melting process occurs on transmission lines overhanging between two towers, ice shedding of one side could simultaneously impact the other side of the lines, causing structural damage to the tower-line system [14,20]. In 2018, multiple accidents of tower collapse and line disconnection occurred in the Hunan power grid, caused by uneven de-icing effects during the large current ice-melting process [11]. Furthermore, de-icing of electrical transmission lines is challenging due to the following aspects:

1. Weather conditions: Extreme cold temperatures and heavy snowfall can make it difficult to access transmission lines and remove ice buildup. Additionally, high winds cause ice to form on transmission lines more quickly and in thicker layers.
2. Line accessibility: Some transmission lines are located in remote or difficult-to-reach areas, making it challenging to perform regular maintenance and de-icing operations.
3. Power outages: If ice buildup causes transmission lines to sag or come into contact with other lines or structures, it can lead to power outages. These outages can be difficult to repair, especially in remote or difficult-to-reach areas.
4. Line damage: Ice buildup can cause damage to transmission lines and associated equipment, such as insulators and conductors. This damage can be costly to repair and can lead to further power outages.
5. Safety concerns: De-icing transmission lines can be a dangerous task, as it often involves working at heights and in inclement weather conditions. Additionally, the use of de-icing chemicals and the operation of de-icing equipment can pose safety risks to workers.
6. Environmental concerns: The use of de-icing chemicals can have negative impacts on the environment.

Therefore, these proposed de-icing schemes are faced with expensive costs, operation difficulties and potential damages in the tower-line system. These techniques above need to be further developed toward more accuracy, safety, lower costs, etc.

Compared with these countermeasures above, surface and coating technology may be an optimal option, considering the operability, expense cost and energy consumption.

Based on different anti-icing mechanisms, several coatings could be roughly classified as electrothermal anti-icing coatings [21], photothermal anti-icing coatings [22], inhibitive anti-icing coatings [23], flexible anti-icing coatings [24], hydrophobic anti-icing coatings, composite anti-icing coatings, etc. Electrothermal anti-icing coatings often refer to semiconductor silicone rubber coatings, mainly employed in the anti-icing insulator [25]. Despite the obvious ice-melting effects, this coating is limited in rime or wet snow weather. Photothermal anti-icing coatings often introduce carbon materials of high heat conductivity, restricted in the unobvious behavior during icing weather [26]. Inhibitive anti-icing coatings aim at the release of freezing point inhibitors, but the inhibitive effect only lasts a short duration. Flexible anti-icing coatings introduce the flexible polymer to achieve low-interfacial toughness [27]. This coating is mainly reported on the surface of insulators, turbine blades, etc. [28]. Hydrophobic or superhydrophobic anti-icing coatings normally refer to the lotus leaf [29], pitcher plants [30], geckoes [31] and penguins [32] in nature [33] (Figure 1). These superhydrophobic coatings have excellent icephobicity, mainly attributed to the delayed freezing of cooled droplets on the surface by reducing the droplet capture and ice adhesion of the interface. Here, Table 1 lists the summary of important superhydrophobic coatings reported for anti-icing protection in the electrical industry.

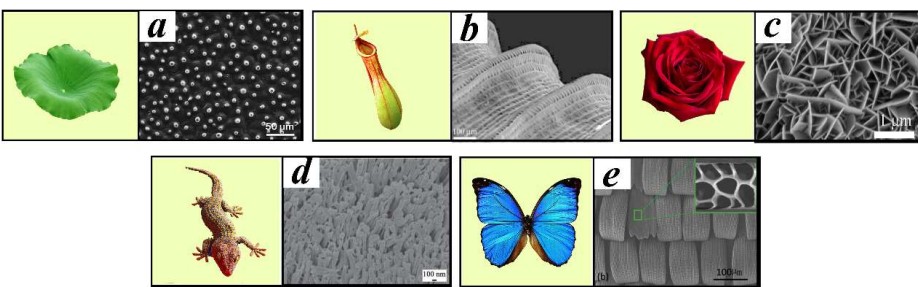

**Figure 1.** Various superhydrophobic surfaces referred to creatures in nature [31,52–54]: (**a**) Lotus leaf [55]; (**b**) Pitcher plants [56]; (**c**) Petals [57]; (**d**) Toes of geckoes [31]; (**e**) Butterfly wings [58].

**Table 1.** Summary of important superhydrophobic coatings reported for anti-icing protection in the electrical industry.

| Preparation Techniques | Modification Agents | Micro-Topography | CA (°) | Anti-Icing Effect | Ref. |
|---|---|---|---|---|---|
| Depositing | RTV SR modified with stearic acid | Micro nanoscale structured roughness surfaces | 150° at −10 °C | Few ice growth spots at a working temperature of −6 °C | [34] |
| Depositing | Silica nanoparticles into polyamide mesh | Controllable meshes with partially embedded nanoparticles | 153.1° | An ice adhesion strength of ~1.9 kPa and a delayed freezing time of ~1048 s | [35] |
| Depositing | MWCNTs mixed with FAS | Hierarchical structure and partially embedded structure | 162.5° | Completely melted with 120 s | [36] |
| Depositing | Doping PVC particles into a silicone matrix | A soft and rigid integrated (SRI) coating | 120°~150° | The ice adhesion of 34.6 kPa when the iced length was 20 cm | [37] |
| Acid etching | Hydrochloric acid-etched surface with FAS | Micro nanoscale holes | 165° | 0.58 kPa at −6 °C | [38] |
| Spray-coating technique | $Al_2O_3$ particles doped in SR solution | Cluster-like structure | 163.4 | 65.4 ± 18 kPa | [39] |
| Spin-coating | Carbon-black, titania or ceria nanopowders in RTV-SR | Grooves between rough asperities | ~150° | Freezing was delayed to ~12–13 min | [40] |

**Table 1.** *Cont.*

| Preparation Techniques | Modification Agents | Micro-Topography | CA (°) | Anti-Icing Effect | Ref. |
|---|---|---|---|---|---|
| Deposition | OD, PF modified with alkylsilane compounds | Micro nanoscale surface | ~160° | Shear stress of ice detachment ($71.5 \pm 15$ kPa) | [41] |
| Anodization and deposition | HMDSO deposited on anodized aluminum | Coral-like nanostructure | ~156° | Keep icephobic properties to 15 icing/de-icing ARF = 2.7 | [42] |
| Deposition | Silicone rubber and alumina nanoparticles | Spongy-like and flower-like structure | 155° | ARF = 1.17 | [43] |
| Spray | Nano $CaCo_3$, silica particles into fluorosilicic and E-51 | Ring-like pits structure | 166.4° | 20% of surface area covered by glaze ice at $-5\,°C$ | [44] |
| Chemical etching | Modified with PFPE | Micro/nanostructures | 160° | Delay the freezing time of water droplets to 5100 s | [45] |
| Salt etching | Modified with silane | Micro/nanostructure | 161.9 | 53% of the surface remained unfrozen in glaze ice after 50 min | [46] |
| RF magnetron sputtering | Sputtered by Zn target, modified with FAS-17 | Nanorods structure | 160° | Frost formation was delayed for 140 min at $-10\,°C$, no degradation after 30 cycles of frosting/defrosting process | [47] |
| RF plasma-sputtering and anodization | PTFE or Teflon sputtered on anodized Al alloys | Nest-like micro nanostructure | 165° | low variation in ice adhesion strength after 15 icing/de-icing cycles | [48] |
| Laser ablation | Modified with methoxy silane | Micro-channel pattern | 169.9° | Superhydrophobicity withstands thermal aging, thermal cycling, UV exposure, long-term ambient outdoor environment exposure and corona exposure | [49] |
| Laser ablation | Modified with FAS-13 and PDMS | Micro pattern | ~145° | Ice adhesion of 60 kPa | [50] |
| Anodization and SLIPS | FAS and silicone oil | Nano-pores structure | 150.3° | Low ice adhesion withstands 150 icing/de-icing cycle | [51] |

However, these superhydrophobic coatings could increase the weight of the substrate, and also face durability and stability under repeated aging, abrasion, corrosion, and corona damages, etc. Here, superhydrophobic surfaces are proposed by combining the rough micro-nano structure and modification agents with low surface energy. Compared with superhydrophobic coatings, superhydrophobic surfaces affect the operating performance (weight and electrical properties) less, and maintain a robust structure for durability and stability [49]. Consequently, techniques of superhydrophobic surfaces show improvements in the application of superhydrophobic coatings and have the profound potential for anti-icing protection of overhead transmission lines.

Here, this review introduces superhydrophobic surfaces in several parts including the theoretical mechanisms of water repellency and anti-icing effects, influencing factors of anti-icing properties, and various preparation techniques to discuss and present the challenges and outlook of superhydrophobic surfaces in the anti-icing protection of overhead transmission line.

## 2. Anti-Icing Mechanism of the Superhydrophobic Surface

Anti-icing mechanisms of superhydrophobic surfaces are mainly reflected in water repellency and icephobic properties. The introduction of theoretical models could lead to insights into the optimized superhydrophobic micro-structure.

### 2.1. Wettability and Hydrophobicity

The wettability of a surface can directly reflect its hydrophobicity. The related parameters [59–61] are included as: the contact angle (CA), contact angle hysteresis (CAH), sliding angle (SA), apparent contact angle (APCA), advancing contact angle (ACA), receding contact angle (RCA), etc. When CA > 150° or CAH < 10°, a surface is generally determined

as a superhydrophobic surface [62]. The mechanism of trapped water droplets on a superhydrophobic surface was first reported by Thomas Young as the equilibrium of gas, liquid and solid phases on a relatively homogeneous surface (Figure 2a) related to surface tensions (interfacial free energies), in the equation of Young [63]. However, this equation is only applicable to relatively smooth surfaces and ignores unideal surfaces with rough microstructures [62]. In addition, the Wenzel model was proposed to modify Young's equation. The Wenzel model in Figure 2b shows the complete wetting of liquid into the rough solid microstructure [64]. The Wenzel model introduced surface roughness to magnify the wetting effect, but hardly demonstrated the water repellency of hydrophobic surfaces with rough microstructures [65]. Moreover, Cassie and Baxter explained heterogeneous wetting behavior in that liquid droplets were held up by the air cushions on the rough microstructures without penetrating the solid phases (Figure 2c) [66,67]. The Cassie-Baxter model introduced the contact interfaces of gas-liquid and liquid-solid, distinguishing between the apparent contact angle (APCA) and actual contact angle (CA) [5,59,66]. In all, the three typical theoretical models can evaluate different wetting behaviors, further indicating that the stabilized and improved water repellency of superhydrophobic surfaces can be achieved by maintaining the air cushion and minimizing the contact fraction of the solid surface [68]. Based on this, water adhesion and motion behavior of different morphological superhydrophobic structures have been studied by many researchers [68–70]. For instance, Liu et al. [31] reported better water repellency of the micro-structures with minimized space, which refers to the morphology of a gecko's toes. Therefore, the designed microstructure of superhydrophobic surfaces significantly influences its water repellency, resulting in the bouncing behavior before freezing.

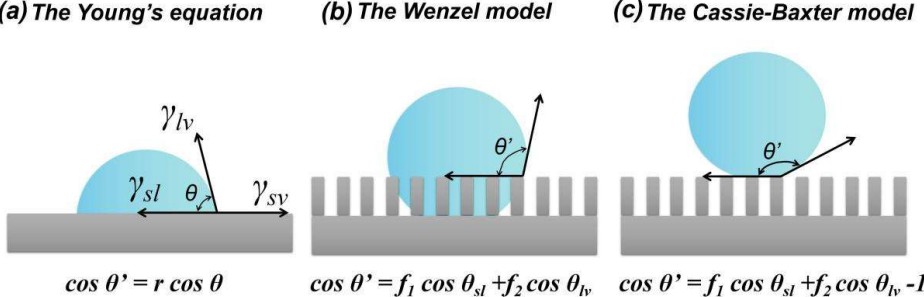

**(a)** *The Young's equation* **(b)** *The Wenzel model* **(c)** *The Cassie-Baxter model*

$$\cos\theta' = r\cos\theta \qquad \cos\theta' = f_1\cos\theta_{sl} + f_2\cos\theta_{lv} \qquad \cos\theta' = f_1\cos\theta_{sl} + f_2\cos\theta_{lv} - 1$$

**Figure 2.** Three typical wetting models: (**a**) Young's equation; (**b**) The Wenzel model; (**c**) The Cassie-Baxter model.

### 2.2. Icing and Icephobicity

Generally, hydrophobicity incompletely corresponds to icephobicity. Water repellency reveals the motion behavior of falling water droplets without remaining on the surface [59,71]. However, the inevitable icing process also involves other mechanisms including the nucleation and growth of ice crystals [61]. The following contents briefly introduce the mechanism of ice freezing.

The source of nucleated ice crystals can be divided into two physical phases of $H_2O$. The water vapor and liquid water in supersaturated or supercooling states are unstable products with higher entropy, then nucleate to ice crystals by overcoming the energy barrier ($\Delta G$) [72,73]. In particular, two forms of ice nucleation were widely reported: uniform nucleation and heterogeneous nucleation. For both forms of nucleation, all ice crystals could experience the ice embryo process until exceeding critical size [74]. During uniform nucleation, all ice crystal nuclei occur uniformly only under the very ideal conditions. Moreover, the uniform nucleation process was reported to be linked to the interactions, connections and rearrangements of hydrogen bonds [73]. With a prolonged time, the nucleated ice crystals could accumulate slowly with the increase of the free energy of the ice-liquid interface.

Generally, heterogeneous nucleation is the more common form due to the presence of foreign introduced particles or solid substrates [75,76]. The icing crystals tend to nucleate onto these heterogeneous surfaces and continuously grow up [77]. For the icephobicity of superhydrophobic surfaces, some researchers reported that the presence of air cushions held the droplets, leading to reducing the contact and heat transfer between droplets and solids. Similarly, some reports studied the mechanism whereby the presence of air cushions could cause ice to nucleate at the triple-phase contact line (vapor/solid, vapor/liquid and liquid/solid) by reducing the heterogeneous ice nucleation barrier [61]. In particular, some researchers studied the micro-patterned surface and found that the similar size of the critical ice nucleus can control the formation of ice crystals [78,79]. Moreover, scholars found that geometric structures with smaller sizes and higher CA could have a higher energy barrier, leading to a lower icing point and a delayed icing process [80]. Regardless of the icephobicity of the superhydrophobic surface before icing, the low ice adhesion of the superhydrophobic surface after icing was also widely reported, due to the partially retained Cassie-Baxter wetting state [81,82]. Yuan et al. have studied the ultra-low ice adhesion of superhydrophobic surfaces after icing, compared with that of the Al substrate [51]. When ice accretion occurs on superhydrophobic surfaces, ice adhesion can be easily overcome only by small external forces or even the gravity of hanging ice ridges. Although the icing mechanism has not been fully clarified, the icephobicity of superhydrophobic surfaces has been widely validated in delaying heterogeneous ice nucleation and growth and decreasing ice adhesion. Nevertheless, the bad durability and aging effect of a superhydrophobic surface could damage the Cassie-Baxter wetting state, resulting in the deterioration of icephobicity, and even the accelerated icing process [82,83]. Therefore, optimizing and maintaining the anti-icing properties of superhydrophobic surfaces has always been a research hotspot for scholars and engineers. Studies on influencing factors of anti-icing properties need to be introduced.

## 3. Influencing Factors of Anti-Icing Properties

Superhydrophobic surfaces have been reported as having good icephobicity, but bad durability and aging effects could be their main disadvantages. Moreover, superhydrophobic surfaces on transmission lines are not entirely consistent with ordinary Al surfaces, as reflected in the intersection of multiple disciplines including hydrodynamics, meteorology, thermodynamics, materials science, electrical engineering, etc. In addition, the anti-icing properties of superhydrophobic surfaces on transmission lines are related to many factors such as: the micro-morphology, micro-climate, the transmission line itself, etc. Therefore, the influencing factors of anti-icing properties need to be evaluated comprehensively. Firstly, the unique icing forms of transmission lines are discussed below.

### 3.1. Icing Forms

The icing forms of transmission lines exhibit unique characteristics. The principal properties of icing on transmission lines [84–91] have been sorted and listed in Table 2. In particular, these given parameters are referred to in previous studies and provide a rough range considering the error of actual measurements. In particular, it is normally accepted in engineering that icing occurs when humidity is higher than 85% RH. In addition, lower temperatures of water vapor tend to bring dryness [92]. An extreme cold environment may hardly be iced. For accuracy, the liquid water content in the air (LWC) is introduced to replace humidity for the discussion of different icing forms. Generally, the three typical icing types on transmission lines are distinguished as atmospheric icing, precipitation icing and frosting [93], depending on environment temperatures, wind speed, LWC, MVD, etc. Atmospheric icing (in-cloud icing) is thought of as ice accretions in dry or wet regimes (Figure 3). Soft and hard rime in a dry regime can be simply distinguished in its density, shape and appearance. Soft rime is thin ice with low MVD and LWC, which is white and opaque in feathery and granular shapes. Hard rime has a higher density (high MVD and LWC) [94] and is in the shape of opaque and eccentric pennants formed by the wind. Glaze

in a wet regime is normally transparent and smooth in cylindrical icicle shapes without air bubbles [84,85]. The glaze layer could grow to the glaze icicles with high water-flux [93], which greatly increases the threat of unsafe operation including flashover of insulators, corona loss, and even line gallop. As a result, glaze and hard rime have much higher adhesion than soft rime, which is difficult to remove. Esmeryan et al. [95] have developed a strategy for atmospheric icing prevention based on chemically functionalized icephobic carbon soot, showing good icephobicity in harsh operational conditions. Moreover, Sharifi et al. [96] combined electrical heaters and superhydrophobic coatings for the protection of atmospheric icing on aircraft. Although many studies of coating technology have been reported for atmospheric icing (in-cloud icing), the feasibility and durability for its large -scale application on transmission lines still need further verification.

**Table 2.** Principle properties of different icing forms on transmission lines.

| Icing Types | | Environment Temperatures (T/°C) | Wind Speed (v/m·s$^{-1}$) | MVD (μm) | LWC (g/m$^3$) | Density (g/m$^3$) | Ice Adhesion |
|---|---|---|---|---|---|---|---|
| Atmospheric icing (in-cloud icing) | Glaze | −5~0 | 1~10 | 1~20 | 0.05~0.6 | 0.1~0.3 | Strong |
| | Hard rime | −15~−3 | 1~15 | 5~35 | 0.4~2.0 | 0.15~0.19 | Strong |
| | Soft rime | −25~−5 | 3~20 | 10~80 | 0.6~3.0 | <0.6 | Weak |
| Precipitation icing | Freezing rain | −3~0 | 2~4 | 300~5000 | 0.11~2.5 | 0.2~0.6 | Strong |
| | Wet snow | −2~3 | 0~9 | 100~300 | 0.02~0.3 | 0.4~0.6 | Initially weak, subsequently enhanced |
| Frosting | Hoarfrost | −5~0 | low | / | / | <0.3 | Strong |

Where MVD refers to the median volume diameter of water droplets, LWC refers to the liquid water content in the air.

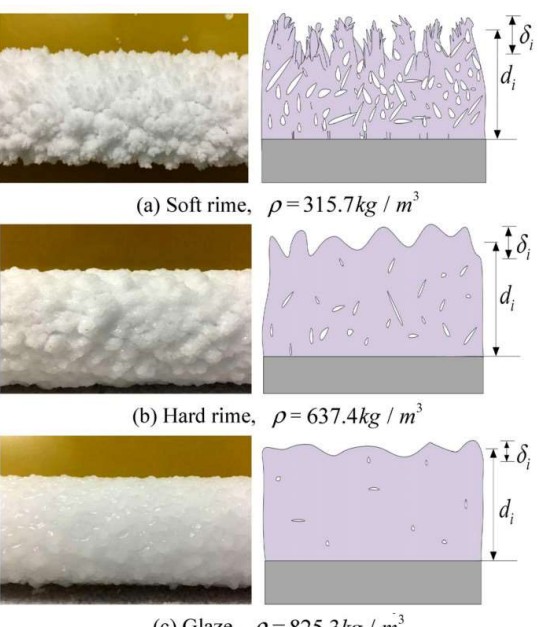

(a) Soft rime, $\rho = 315.7 kg/m^3$

(b) Hard rime, $\rho = 637.4 kg/m^3$

(c) Glaze, $\rho = 825.3 kg/m^3$

**Figure 3.** Morphology of typical atmospheric icing forms [85]: (**a**) Soft rime; (**b**) Hard rime; (**c**) Glaze.

In addition, precipitation icing is another common icing form and is dangerous to the safety of transmission lines. Precipitation icing includes freezing rain [97,98] and wet snow [89,90], mainly differing in precipitation forms of snow or rain. Freezing rain refers to when rain falls and sticks on a super-cooled surface, then ices immediately or after twisting on conductors. This icing form may be considered an enhanced glaze, which has high MVD, LWC and ice adhesion. Wet snow [89,90] is mainly falling snow that sticks on a cooled surface then keeps accumulating. Ice adhesion is weak initially, then subsequently enhanced. Nevertheless, precipitation icing frequently occurs under serious

weather conditions, resulting in excessive ice accretions. In recent years, Chen et al. [98] studied the drop size distribution of freezing precipitation (freezing rain or drizzle) and concluded the formation mechanisms. Moreover, Mohammadian et al. [89] studied a prediction model of wet snow shedding from overhead structures. Nevertheless, studies of prediction models and prevention techniques on precipitation icing need to be further improved.

Frosting (hoarfrost) is the third familiar icing form with strong adhesion, and often occurs after the sublimation of water vapor under the low wind. Normally, hoarfrost is hard to remove for high ice adhesion. Moallem et al. [99] studied experimental measurements of hydrophobic coatings on frosting performance and found an obvious improvement in heat transfer capacity. Additionally, Varanasi et al. [82] found that frost formation could significantly compromise the icephobic properties of superhydrophobic surfaces. Therefore, frosting is the important icing form resulting in the failure of anti-icing superhydrophobic surfaces. Recently, Esmeryan et al. [100] reported the icephobic properties and fabrication scalability of carbon soot for anti-frosting applications. The composition and structure design could be the research focus of superhydrophobic surfaces for its anti-frosting effects.

Furthermore, Rønneberg et al. [101] compared three ice types including precipitation ice, in-cloud ice and bulk water ice on the same aluminum substrate and found the ice adhesion strength inversely correlates with the density of ice. In the actual measurement of ice accretion, all the above icing types shall be considered.

Moreover, icing types can significantly affect the icing morphologies of conductors. Considering the unique conductor surface and icing environment factors [102], the macromorphology of iced conductors is observed in different shapes. Here, typical ice morphology profiles of the iced conductors in Figure 4 [103] are included as: (a) crescent shape; (b) wing shape; (c) wing and eccentric circle shape; (d) eccentric oval shape; (e) eccentric round shape; and (f) wing and round shape. These icing macro-morphologies show the actual results of a transmission line in the Liupanshui and Xuefeng mountains. Crescent- and wing-shaped icing morphologies (a, b) are the typical morphologies during the initial icing, while other morphologies show irregular shapes after uneven icing accretions. Many scholars studied icing growth and covering characteristics according to these typical models [88,102]. Moreover, the influencing factors are the focus of scholars to study and evaluate icing growth models and anti-icing performance. These theoretical formulas and models are conducive to handling the actual requirements of line anti-icing.

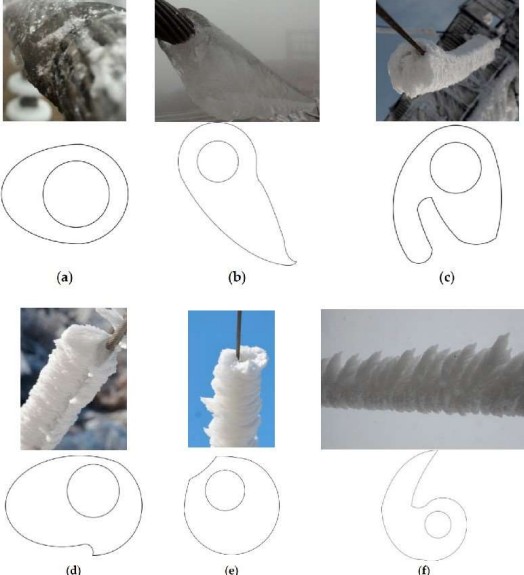

**Figure 4.** Typical macroscopic ice shape profiles of iced conductors [103]: (**a**) Crescent shape; (**b**) Wing shape; (**c**) Wing and eccentric circle shape; (**d**) Eccentric oval shape; (**e**) Eccentric round shape; (**f**) Wing and round shape.

### 3.2. External Factors

Influencing factors on anti-icing properties are divided into external factors and internal factors. External factors mainly refer to the surrounding environment of transmission lines. Traditional impact factors of superhydrophobic surfaces include temperature and humidity. Generally, low temperature and high humidity conditions make water vapor condensate indiscriminately in the gaps of microstructures (Figure 5), partially destructing the Cassie-Baxter state and losing the superhydrophobicity [79,82].

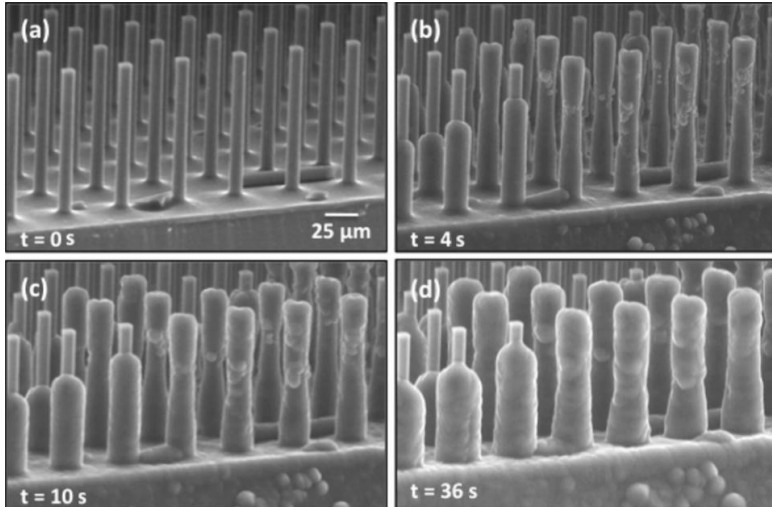

**Figure 5.** A superhydrophobic structure under low temperature and high humidity [82]. (**a**) Superhydrophobic surface; (**b**–**d**) Superhydrophobic surface after frosting for different time.

Nevertheless, the unique operating environments of conductors [104] also include wind velocities [94,105–107], wind direction [106,107], high temperatures, ultraviolet exposure, contamination deposits, corrosion, mechanical wear and corona loss. Generally, wind speed is inversely proportional to temperature, as reflected in the icing thickness of a bare conductor [105], even resulting in the line galloping [106,107]. Yu et al. [108] found that ice shedding occurred on the 20 mm icing film of a conductor (ground) wire in the first 20 s at wind speeds of 15 m/s. In addition, Kermani et al. [106] simulated the fracture behavior of atmospheric ice on bare conductors under different wind speeds and found it sometimes occurred during line galloping when stresses arose at a wind speed of 5 m/s. However, high wind speed could be conducive to the anti-icing behavior of superhydrophobic surfaces compared with Al substrates due to the bouncing behavior of water droplets and low ice adhesion [94,105]. Xue et al. [105] studied the anti-icing properties of superhydrophobic electrothermal film under different temperatures and wind speeds and concluded the main factors to be water collection rate per unit area and wetting coefficient. Belaud et al. [94] studied superhydrophobic aluminum oxide surfaces for aeronautic applications and found low atmospheric ice adhesion under wind tunnel tests. Wind direction mainly reflects on the windward and leeward sides of conductors in windy and less windy regions. A thicker ice layer is normally observed on the windward side in windy regions, compared with the leeward side in less windy regions [88,103].

Based on the above, excessive ice accretions may hardly occur on a superhydrophobic surface, but durability is a technical problem to overcome. Other environmental factors are the main challenges to the durability and stability of a superhydrophobic surface. High temperatures refer to the operating temperatures of a transmission line influencing the aging effects, which are generally in the range of 60~100 °C. CIGRE [49] reported temperature cycling tests that increase the temperature to 60 °C for 4 h and keep it for 8 h, then lower the temperature to −20 °C for 4 h and keep it for 8 h. Testing temperatures can also be adjusted for the actual line temperature (maximum value) and local climates (minimum value). In particular, many scholars have found the self-healing effects of anti-icing properties of

superhydrophobic surfaces under high temperatures, which also indicate good durability [17,51,109]. In addition, ultraviolet exposure is also a slow aging factor resulting in the deteriorated superhydrophobic properties of coatings [110]. Moreover, contamination deposits are involved in pollutants and sand dust, as reflected in the pollution flashover of insulators. The self-cleaning properties of a superhydrophobic surface can mitigate surface salt deposit density to some extent. For instance, Xu et al. [111] prepared photoinduced TiO$_2$ coatings by decomposing the contamination under the catalysis of ultraviolet light. Corrosion, which is reflected in the salt corrosion in coastal areas and the galvanic corrosion between the aluminum line and the galvanized steel core of an aluminum conductor steel reinforced (ACSR) in Figure 6 [112]. Although superhydrophobic coatings are widely reported as good corrosion protection for hydrophobicity, it is also noted that the mechanical wear of coatings under the galloping of lines could also damage the coatings and accelerate localized corrosion.

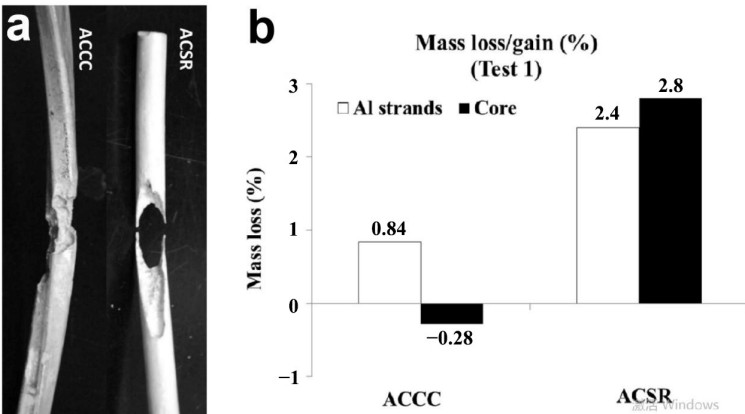

**Figure 6.** The galvanized corrosion of bare ACSR and superhydrophobic coated ACSR [112]: (**a**) The macroscopic corrosion morphologies. (**b**) The corresponding corrosion mass loss.

　　　　Corona damage could be the most aggressive and destructive factor influencing the durability of superhydrophobicity [49]. In recent years, the delivered voltage level increases for the development of the power grid. Corona discharge is inevitably generated on the conductor surface for high and heterogeneous electric field strength [113,114]. Electric field distortion is caused by dropped water, accreted icing, surface defects, etc. [115,116]. Wang et al. [117] have found the transition from hydrophobicity to hydrophilicity on superhydrophobic coatings after corona treatment (Figure 7a,b). A similar phenomenon was also reported on the RTV coatings of insulators [118,119]. Nevertheless, the corona discharge of superhydrophobic surface conductors is not completely consistent with the needle-plate experiment on insulators [120]. Megala et al. [121] prepared thick coatings of high dielectric constant to reduce corona loss (Figure 7c). Zhang et al. [122] have found improved corona performance of superhydrophobic coated conductors (Figure 7d,e) attributed to the repellency of water droplets. In all, superhydrophobic coatings may improve corona performance, but the anti-icing properties will age inevitably under high corona discharge. Corona discharge could probably be the most challenging factor of the durable anti-icing properties of superhydrophobic surfaces.

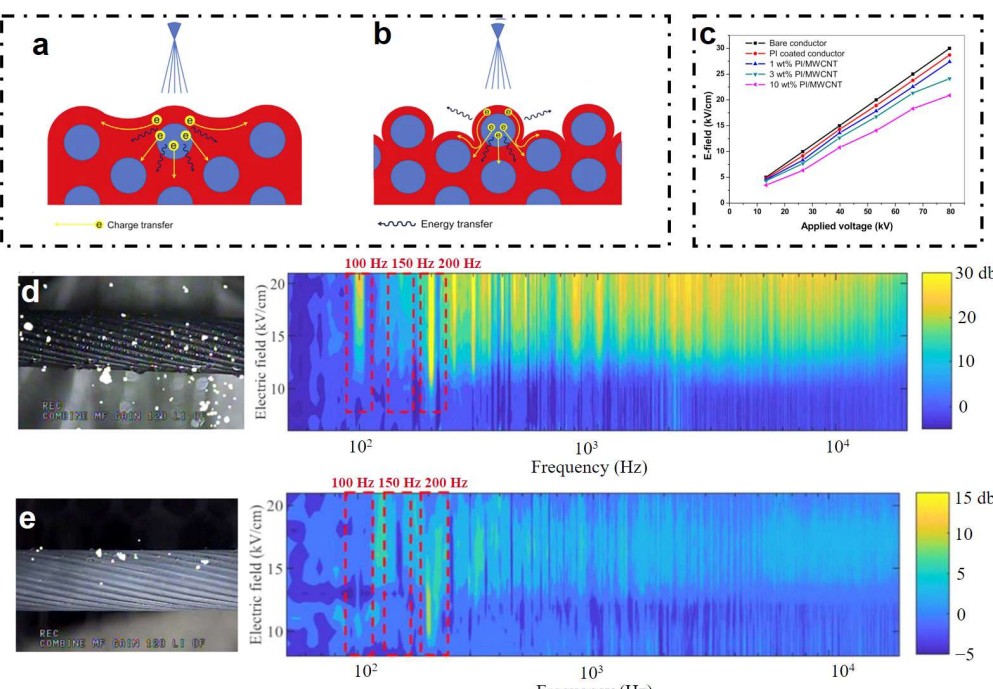

**Figure 7.** Corona performance of superhydrophobic coatings: (**a**,**b**) The corona discharge mechanisms of the bare substrate and micro nanostructure [94]. (**c**) The inception corona voltages of coated and bare conductors [121]. (**d**,**e**) The UV images and audible noise of bare conductors and superhydrophobic coated conductors [122].

### 3.3. Internal Factors

Internal factors of the anti-icing properties of superhydrophobic coatings include rough microstructure, comprehensive properties, durability, the preparation technique of coatings on the curved conductors, etc. Firstly, the designed rough morphology of a superhydrophobic surface crucially affects its anti-icing properties, which is reflected in the wetting state of droplets, the heat transfer reaction and the ice nucleation process [4,60]. The different morphologies (Figure 8) include micro-ratchets, micro-patterned [123], multi-peaks, nanorods, nanoneedles [70], flower-like, nanopores, micro-meshes [124], tilted nanorod, cactus-like ball structure [125], etc. These designed superhydrophobic surfaces show their respective advantages but a similar structural feature of high roughness and ordered microstructures. In particular, not all superhydrophobic micro-topography can exhibit optimal anti-icing properties. Under low temperature and high humidity conditions, some micro-structures cannot hinder the nucleation and condensation of cooled water vapor, which is reflected in the interlocking of the iced structure [79,82]. Zhang et al. [126] studied droplet condensation on superhydrophobic nanoarrays using lattice Boltzmann modeling and found that nanostructures with taller posts and a high ratio of post-height-to-spacing will be beneficial to anti-condensation superhydrophobic properties. In addition, Wang et al. [127] found that higher micro-cone heights and lower micro-cone pitches of rich micro-nano structures can improve the robustness of the superhydrophobic state. Moreover, each raised pattern of these micro-structures could have the critical size of optimal anti-icing performance [5], as reflected in the large storage of low surface energy agents, good water repellency, low ice adhesion and long delayed icing time. Therefore, the structural design of a rough superhydrophobic surface is of significance to its anti-icing properties.

Moreover, the preparation of superhydrophobic surfaces should include chemical functionalization on the rough topographical surface. Normally, low surface energy materials are adopted to modify the as-prepared surface for superhydrophobicity. The modification of these materials aims to modulate with chemical functional groups, including hydroxy and other hydrophilic groups [128]. Here, long-chain organic silanes, fatty acids and

fluorinated compounds are the most common modifications [129]. Zhang et al. [130] studied stoichiometrically controlled reactions of OTS-based hexane/water and confirmed the importance of the modifiers ratio. In addition, Zhizhchenko et al. [131] found that ODTMS layers on a metal surface get hydrolyzed gradually and degrade slowly when interacting with water molecules. However, Xiang et al. [132] found that pore size and porosity are key factors for superhydrophobic surfaces to obtain durable anti-icing properties, attributed to the large storage of lubricant, reduced lubricant loss and accelerated self-healing. Therefore, the structure design of rough topographical surfaces and the capacity volume of modified materials could affect the durability of anti-icing superhydrophobic surfaces. Furthermore, lubricants have been recently introduced to superhydrophobic surfaces for the construction of liquid-infused surfaces. Numerous viscous oils are adopted, classified as fluorous lubricants, including Fluorinert, Krytox oils, etc., and environmental nonfluorinated hydrophobic lubricants, including silicone, hydrocarbon oils, etc. [2].

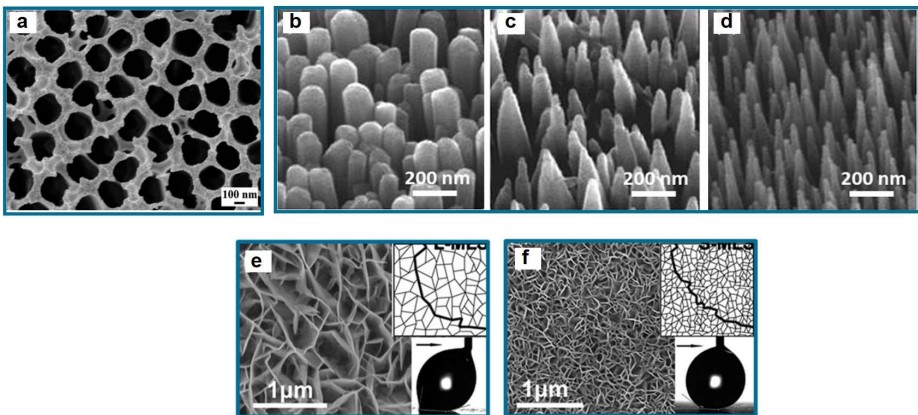

**Figure 8.** Different optimized morphologies of superhydrophobic surface: (**a**) Nanopores structure [31]; (**b**) Nanorods structure; (**c**) Nanopencils structure; (**d**) Nanoneedles structure [70]; (**e**,**f**) Mesh-like structure [124].

Related properties of superhydrophobic coatings include mechanical, corrosion, electric, and thermal properties. The mechanical properties are reflected in surface stress and adhesive force. High surface stress can repeatedly withstand the scratch and fracture of coatings [133], especially within the overhead line environment. Moreover, multiple occurrences of icing and de-icing cycles also require surface hardness to avoid the destruction of the microstructure. In addition, corrosion properties require the integrity and protection of superhydrophobic coatings to hinder the erosion of corrosive products (salt fog, acid rain and industrial pollutants) [134]. Moreover, electric properties could be another important factor for deep consideration. Electric properties include resistivity, dielectric properties and corona performance. The resistivity and dielectric properties of coatings reflect heat loss for the skin effect under the transmission of conductor lines [135]. Normally, appropriate thickness and dielectric loss of coatings can effectively reduce line loss [136]. Corona performance is the key performance for coated conductors, including corona inception voltage, audible noise (AN), radio interference (RI) and corona loss (CL). Recently, superhydrophobic coatings were reported to enhance corona performance by reducing electric field distortion for the attached water droplets. Nevertheless, some scholars also found a negative effect of high surface roughness on corona performance. The affecting mechanism of superhydrophobic coatings on corona performance needs to be further studied. In addition, thermal properties mainly include heat resistance and thermal conductivity. Heat resistance requires aging resistance under the high operating temperatures (~100 °C) of the transmission conductor surface. Thermal conductivity was reflected not only in heat dissipation during operation, but also the quick response and high efficiency of the ice-melting effects [85]. Good thermal conductivity is also an important component of an anti-icing property.

The durability of superhydrophobic coatings generally aims at external factors, which are introduced above [137,138]. Correspondingly, researchers tried various methods to estimate and measure the stability and durability of superhydrophobic coatings, including sand erosion, rubbing, water jetting, sandpaper scratching, UV weathering, etc. Although superhydrophobic coatings inevitably encounter the aging process, improved superhydrophobic Al surfaces have been proven to have robust durability and partial self-healing properties for the composition of inorganic ceramics and storage of modifiers. Lian et al. [49] studied the long-term durability of superhydrophobic surfaces under natural ambient outdoor conditions for 1 year. Yuan et al. [51] studied the self-healing behavior of SLIPS superhydrophobic surfaces and found durable anti-icing properties that withstand ~150 icing/de-icing cycles via 5 times self-repair for 1 h and 100 abrasion cycles via 4 times self-healing for 1 h (Figure 9). The reported self-healing property was attributed to the modification agents migrating to the surface under the capillary force of the microstructure, especially at a high temperature. However, the self-healing behavior under the actual natural environment and service conditions still needs to be verified. In particular, various similarly advanced preparation techniques such as laser ablation may not be easily applied to the geometrically complex surfaces of an aluminum conductor steel reinforced (ACSR). In recent years, Liao et al. [46] have prepared an anti-icing conductor with a superhydrophobic surface using the chemical etching technique. Nevertheless, alternative reliable methods to widely prepare superhydrophobic surfaces on conductors have not been reported. Furthermore, the preparation technique of coatings on curved conductors causes application difficulty for various superhydrophobic surfaces. Therefore, this review introduces the preparation methods of superhydrophobic surfaces in recent years for other possible alternative options.

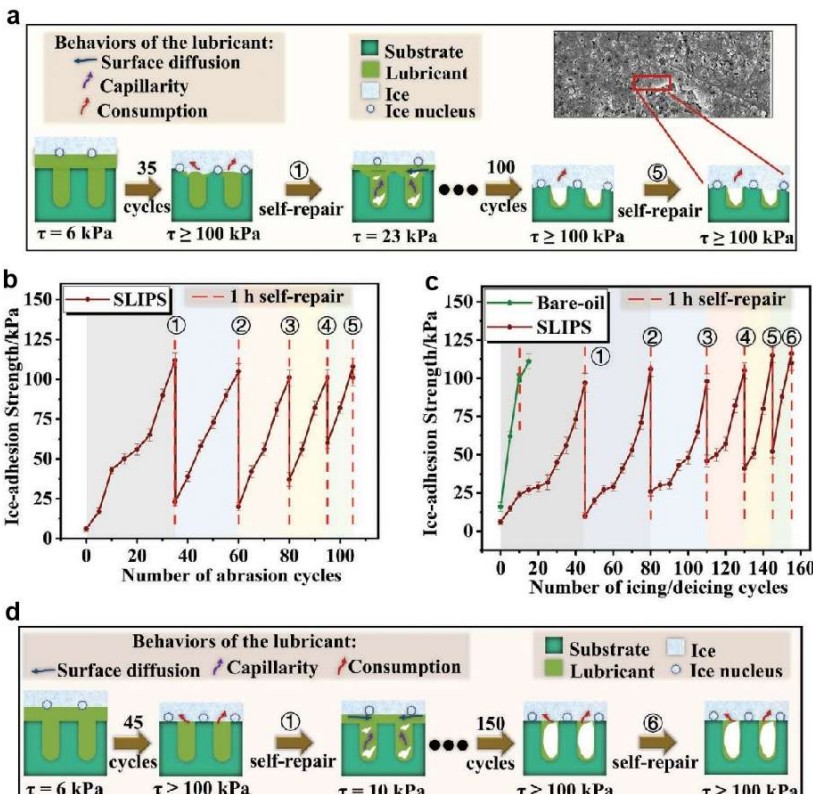

**Figure 9.** The self-healing mechanisms and results of SLIPS surface after: (**a,b**) Artificial scratches and (**c,d**) icing–de-icing cycles [51].

## 4. Preparation Techniques for Superhydrophobic Surfaces

To date, numerous synthetic techniques for preparing superhydrophobic surfaces on Al substrates have been reported. The construction of a superhydrophobic surface involves preparing the rough micro or nanostructure, followed by the modification of the low surface energy material. The designed microstructure of these superhydrophobic surfaces is similar to the lotus leaf, pitcher plants, geckos and penguins in nature. The main reported preparation techniques include liquid-phase deposition, chemical etching, anodic oxidation, laser etching, mechanical templating, etc. In particular, mechanical templating could not prepare the nanostructure. The random micron-scale structure could only be used as a pretreatment. Physical and chemical vapor deposition techniques could hardly be realized on conductors for requiring the vacuum chamber. Therefore, the following contents show the progress of potential application techniques on transmission conductors.

### 4.1. Liquid-Phase Deposition

Liquid-phase deposition is the traditional preparation technique, including dip-coating [130], spray-coating [139,140], etc. Several traditional coating technologies can be introduced as follows:

1.  Spraying: A superhydrophobic coating can be sprayed onto the surface of transmission lines using specialized equipment. This method is fast, efficient, and can provide a uniform coating.
2.  Painting: Superhydrophobic coatings can also be applied using a paint brush or roller. This method is more labor-intensive but can be useful for coating irregular or hard-to-reach surfaces.
3.  Dip coating: In this method, transmission lines are dipped into a solution containing the coating material and, after that, it is dried.

Here, one-step spraying technology in Figure 10 a is widely utilized for simplicity and convenience. This method is not only specific to the plane but geometrically complex surfaces including curved conductors. The recent advance in the one-step spraying method is focused on the composition of spraying materials [141]. Widely applied raw spraying materials include silicone rubber (SR) [8], room-temperature vulcanized silicone rubber (RTV) [142], epoxy resins, esters, stearate, aliphatic acid, polytetrafluoroethylene (PTFE), etc. In recent years, more flexible polymers were introduced to improve interlayer adhesion and mechanical performance of sprayed coatings such as polydimethylsiloxane (PDMS) [24], polydopamine (PDA) [130], polyurethane (PU) [61], acrylic resin [4], silane coupling agent (SCA) [27], silicone adhesives [143], etc. Zhang et al. [144] studied the octadecyl trichlorosilane (OTS)-based solution via stoichiometric silanization that can be utilized as dipping or spraying materials for various solid substrates. Moreover, various nanoparticles have been studied as novel raw materials for nanoscale roughness and functional performance, such as Graphene oxide (GO), $SiO_2$ particles [145], HDTMS nanocapsules [130], calcium carbonate [140], ZnO, $Al_2O_3$ powders, etc. These novel polymer coatings have been reported to have good icephobicity and self-healing properties, but relevant aging and electric performance tests on conductors require further verifications. Yang et al. [146] recently proposed low-emissivity solar-assisted superhydrophobic (LE-SS) coating on power line conductors, composed of novel TiN and $SiO_2$ nanoparticles (Figure 10 b). In their study, although excellent anti-icing and solar de-icing properties have been reported, the related anti-aging and electric properties may still be studied. Therefore, validation of durability remains a challenge to these sprayed coatings on conductors.

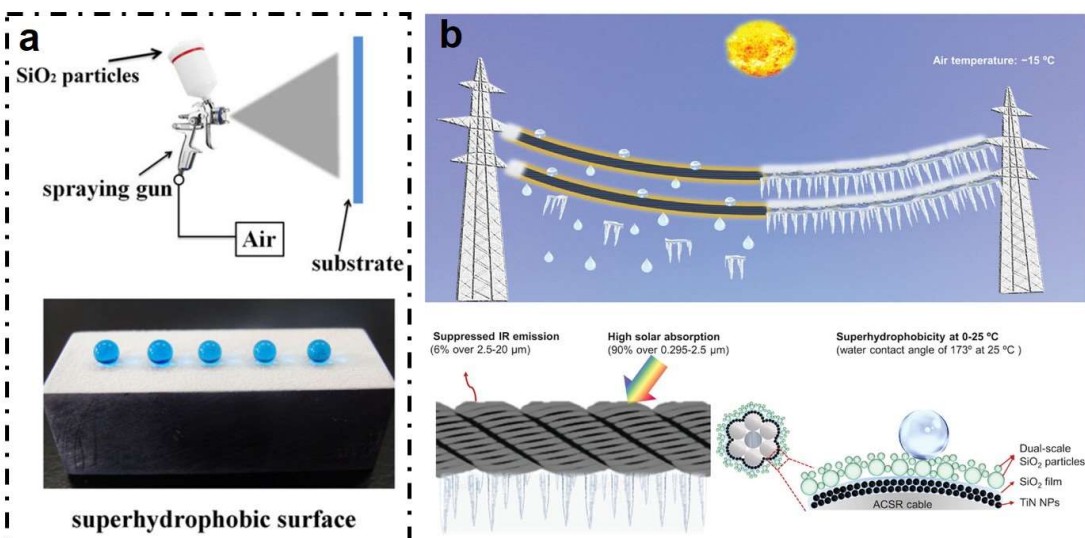

**Figure 10.** The one-step spraying technique: (**a**) Preparation process [139]. (**b**) Application of the low-emissivity solar-assisted superhydrophobic (LE-SS) coating on transmission lines [146].

### 4.2. Chemical Etching

Chemical etching as a convenient and low-cost technique is widely applied on the complex surfaces of conductors. Chemical etching can be realized in numerous etching solutions: acid etching, salt etching, alkaline etching, hot water, etc.

Acid etching is leaving the Al-based specimens immersed in an acid solution for a certain time, then modifying them. Xiao et al. [147] studied superhydrophobic Al surfaces by HCl etching and thiol–ene coupling. Similarly, Yang et al. [148] reported a anti-icing/frosting superhydrophobic Al surface with hydrangea-like microstructures, after being etched in HF and HCl mixed acid solutions and modified by FAS-17. Moreover, Jin et al. [149] successfully prepared superhydrophobic Al conductors with good anti-icing performance using HCl acid etching and stearic acid modification (Figure 11a). However, the limited micro-structural controllability and existing destruction on Al-based surfaces are the main disadvantages. Salt etching involves performing a replacement reaction of Al ions on the surface in a salt solution. Typical copper ion solution is widely utilized to prepare superhydrophobic micro-structures. Liao et al. [46] fabricated a superhydrophobic surface on Al foils by etching in $CuCl_2$/HCl solution and modifying by HDTMS. This prepared surface exhibited a WCA of 161.9° ± 0.5°, delaying the freezing time of water droplets for 475 s. Zuo et al. [150] further fabricated a coral-like superhydrophobic surface using the $CuCl_2$ etching method (Figure 11b) to improve anti-icing properties of the unfrozen water droplet on the prepared surface at −6 °C, which remained for over 110 min. In particular, salt etching should be realized under the dissolution of natural oxide film on a Al-based surface and the replacement reaction of salt ions. Alkaline etching is based on the reaction of an Al-based substrate and alkaline solution and the generation of a rough superhydrophobic structure. Xu et al. [124] prepared a rough superhydrophobic surface on 1060 Al alloys using NaOH etching, which delayed the icing time of droplets at −18 °C for 26 h. Recently, alkaline etching is considered to be combined with other methods such as anodic oxidation [83], sandblasting [123], etc. Shen et al. [123] prepared hierarchical micro-nano structures with low ice adhesion (75 kPa at −10 °C) by combining NaOH etching and sandblasting techniques in Figure 11c. This technique can be widely adopted as an assisted preparation method on an Al surface.

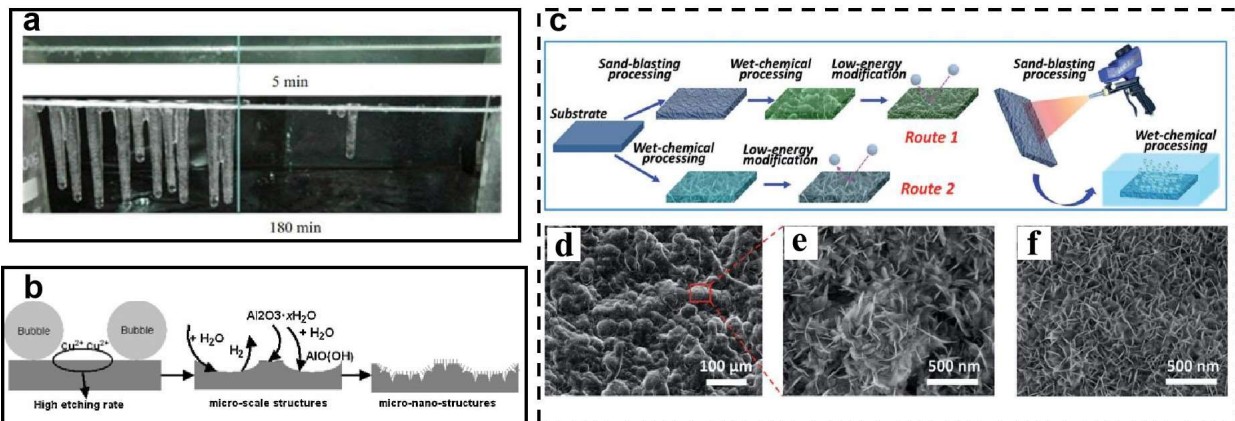

**Figure 11.** Chemical etching technique: (**a**) The anti-icing behavior of the acid etched Al conductors. (**b**) Salt etching mechanism of Al substrate. (**c**) Alkaline etching combined with sandblasting techniques [96,100,119]. (**d**–**f**) The morphology of etched surface.

Hot/boiling water treatment aims at the reaction of aluminate that crystallizes as nanosheet-like hydroxides on a surface. The modifier agent can react with hydroxyl groups of hydroxides for superhydrophobicity [148]. Han et al. [143] prepared an icephobic Al superhydrophobic surface using boiling water treatment and silane modification. The delayed icing time of droplets on this surface exceeds 2 h at −20 °C. In addition, hot/boiling water treatment is usually seen as the further preparation of nanostructures, combined with alkaline etching, salt etching, anodic oxidation, etc. In all, chemical etching shows many advantages, but inhomogeneous micro-construction could potentially result in unstable anti-icing properties. Moreover, chemical etching can combine with other methods to prepare micro-nano structures, which leads to greater advantages.

### 4.3. Anodization Technique

The anodization technique is the traditional processing of an Al surface. The porous anodic oxide film is normally formed under an electrochemical corrosion reaction, which is seen as the ideal superhydrophobic micro-structure for the ordered and regular arrangement of etched pores [30,151]. Boinovich et al. [152] prepared an anodic oxide porous structure on Al wires after the functionalization of fluoro silane. The axial shear adhesive strength reaches about 113 kPa at −8.5 °C, indicating the enhanced anti-icing property. Moreover, Liu et al. [83] prepared a robust and anti-icing superhydrophobic Al surface via anodization and infusion of FAS-17, with excellent self-healing properties that heal the superhydrophobicity after 8 times oxygen plasma or 64 times abrasion under 12 kPa pressure for 7.68 m. Besides low surface energy agents, lubricants are recently chosen to be infused into the porous structure. Normally, these slippery liquid-infused porous surfaces (SLIPS) are inspired by pitcher plants and exhibit good liquid repellency and self-healing properties [51]. Sun et al. [30] perfused silicone oil into the porous oxide film modified with fluorosilane in Figure 12 and prepared a lubricating icephobic superhydrophobic Al surface with a low ice adhesion of 13.4 kPa. In conclusion, the anodization technique is another potential technique for application in overhead lines, due to its economical costs and controllability. Combined with chemical etching, infused lubricants and other methods could further improve anti-icing performance. Nevertheless, the durability of self-healing properties and the related electric performance of superhydrophobic anodized conductors should also be explored.

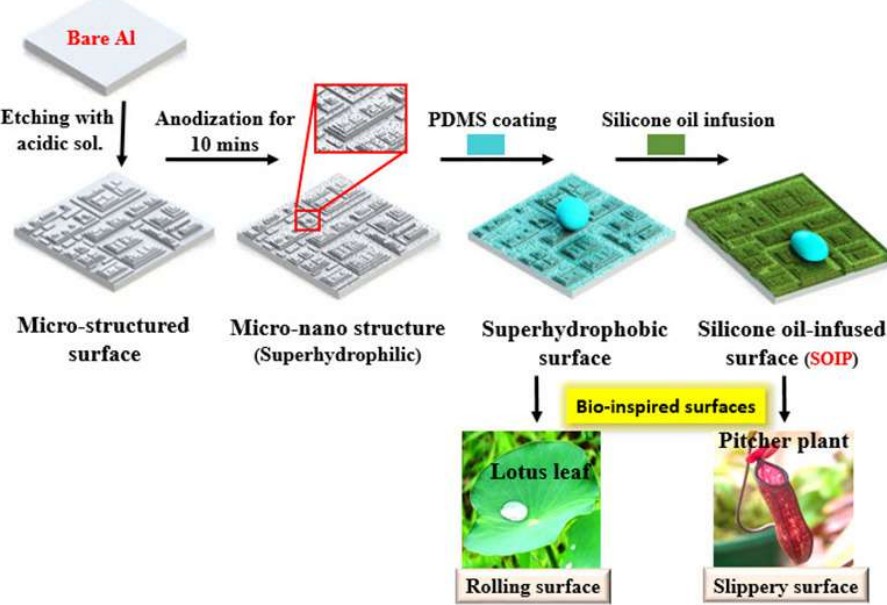

**Figure 12.** The preparation process of SLIPS surface by combining anodization and etching methods [30].

### 4.4. Laser Ablation Technique

The laser ablation technique is a precise machining method using a laser beam that ablates a surface to form micro-patterns or texture topography with grooves [153]. Subsequent modification can endow the surface with anti-icing properties. The technique has advantages in high repetition rate, stability, and not creating pollution. Recently, Wang et al. [127] studied the wetting behaviors of four well-designed micro-nano structured surfaces via ultrafast laser ablation (Figure 13), indicating the precise control of this technique. Li et al. [50] studied the laser-ablating surface of 7075 Al alloys modified with silicon hydride and PDMS, and reported a low ice adhesion of about 60 kPa. Moreover, Lian et al. [49] prepared the micro-textures of laser ablating Al surfaces modified with HDTMS and measured the durability under simulated environments of overhead conductors. The prepared surface showed robust superhydrophobicity under thermal aging, thermal cycling, and UV exposure. In all, the laser ablation technique can accurately control the structural parameters (spacing, size, depth, etc.) of micro-nano structures, which helps the comprehensive analysis of the anti-icing mechanism and improves the optimization of superhydrophobic surfaces [154]. Nevertheless, the technical difficulty of applying on curved and geometrically complex surfaces still needs further study and overcoming.

### 4.5. New Progress in Preparation Techniques

So far, various novel designs of superhydrophobic surfaces have been proposed: soft polymer coatings of low interfacial toughness, hierarchical micro-nano structure [123], SLIPS, etc. Combinations of different superhydrophobic techniques were widely attempted and adopted to exert their advantages. In addition, other anti-icing mechanisms were considered to incorporate into superhydrophobic surfaces, such as icing inhibitors, lubricants, photothermal carbon materials, semi-conductive electrothermal coatings, magnetothermal materials, etc. In all, the preparation and optimization of superhydrophobic surfaces have a promising future in anti-icing protection of transmission lines.

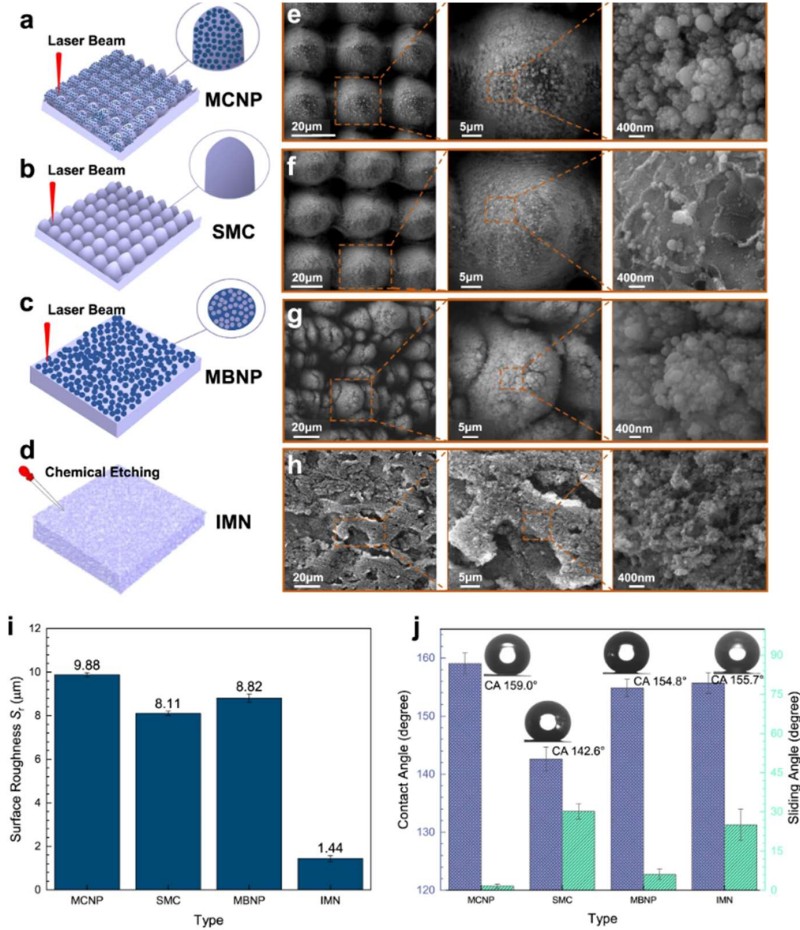

**Figure 13.** Microstructure of laser and chemical etched specimens: (**a**,**e**) Double-scale periodical micro-cones by laser; (**b**,**f**) Single-scale periodical micro-cones by laser; (**c**,**g**) Double-scale random microbumps by laser; (**d**,**h**) Irregular micro-nano structure by chemical etching. Roughness and wettability data of above specimens: (**i**) Comparison of surface roughness; (**j**) Comparison of wettability [127].

## 5. Conclusions and Outlook

Ice accretion on transmission lines is a challenging disaster for power systems. Superhydrophobic coatings have attracted much attention for their significant anti-icing properties, feasibility, simplicity and convenience. Although numerous related advanced studies of superhydrophobic coatings on Al surfaces have been reported, unique operating environments and geometrically complex surfaces of transmission lines restrict the adoption of these superhydrophobic coatings.

This review introduces several parts, summarized as superhydrophobic anti-icing mechanisms, influencing factors of icing on conductors, and potential preparation techniques of superhydrophobic surfaces on transmission lines. From the mechanism of water repellency and anti-icing, the construction and optimization of superhydrophobic microstructures have been discussed. According to the general icing forms and morphologies of overhead transmission lines, related external factors can be further determined. Together with other unique external and internal factors, corresponding measurements of properties on conductors are necessary, including anti-icing properties, durability, surface stress, mechanical properties, corrosion properties, thermal properties, electric properties, etc. Here, anti-icing properties, durability and electric properties are worthy of more attention.

Moreover, the preparation of techniques for curved conductors is another challenge. This review introduces the principle and latest progress of potential application techniques on transmission conductors, including liquid-phase deposition, chemical etching, anodic

oxidation and laser etching techniques. Although these technologies have corresponding shortcomings, the new progress of these techniques provides the developed potential of anti-icing superhydrophobic conductors. In addition, the introduction of various novel anti-icing mechanisms could improve the anti-icing properties of superhydrophobic surfaces.

In summary, superhydrophobic surfaces can be combined with other de-icing methods such as line heating to provide an even more effective solution. Notedly, the applicability and durability of superhydrophobic coatings must be reasonably addressed since they are not a permanent solution, as reflected in the wear-off over time, and require periodical reapplication. Furthermore, the mentioned forwarding and challenges of superhydrophobic surfaces should be focused on to come closer to realizing long-term anti-icing effects on overhead transmission lines.

**Author Contributions:** Conceptualization, X.D. and Y.Y.; validation, B.L., J.B., J.H., C.D., W.C., T.Z. and R.L.; formal analysis, B.L., J.B., J.H., C.D. and R.L.; investigation, X.D.; resources, B.L.; writing—original draft preparation, X.D.; writing—review and editing, X.D.; visualization, B.L. and Y.Y.; supervision, Y.Y. and R.L.; project administration, B.L.; funding acquisition, B.L. and Y.Y. All authors have read and agreed to the published version of the manuscript.

**Funding:** This research was funded by the Electric Power Research Institute of Guizhou Power Grid Co., Ltd., China (Contract No. 0666002022030101HX00001).

**Institutional Review Board Statement:** Not applicable.

**Informed Consent Statement:** Not applicable.

**Data Availability Statement:** The raw/processed data required to reproduce these findings cannot be shared at this time due to legal or ethical reasons.

**Conflicts of Interest:** The authors declare no conflict of interest.

## Abbreviations

Abbreviation list of short designations in this study.

| Designation Categories | Abbreviation |
| --- | --- |
| Room temperature vulcanized silicone rubber | RTV-SR |
| Silicone rubber | SR |
| Depositing | RF |
| Multiwall carbon nanotubes | MWCNTs |
| Polyvinyl chloride | PVC |
| Fluoroalkylsiane | FAS |
| Octadecyltrichlorosilane | OD |
| Perfluorooctyltrichlorosilane | PF |
| Fluoroalkylsioxane | FAS-17 |
| Hexamethyldimethyl ether | HMDSO |
| Aluminum | Al |
| polydimethylsiloxane | PDMS |
| Methoxypropyltrimethoxysilane | PFPE |
| Octadecyltrimethoxysilane | ODTMS |
| Hexadecyltrimethoxy silane | HDTMS |
| Octadecyl trichlorosilane | OTS |
| Polyurethane | PU |
| Graphene oxide | GO |
| Silane coupling agent | SCA |
| Contact angle | CA |
| Contact angle hysteresis | CAH |
| Median volume diameter of water droplets | MVD |
| Liquid water content in the air | LWC |
| Low-emissivity solar-assisted superhydrophobic | LE-SS |
| Adhesion reduction factor | ARF |

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
