# Peer review of "A Review on Superhydrophobic Surface with Anti-Icing Properties in Overhead Transmission Lines"

_coatings, doi:10.3390/coatings13020301_

Round 1

Reviewer 1 Report

See attachment

Reviewer 2 Report

The paper entitled “Improvement of Mechanical Properties and Solvent Resistance  of Polyurethane Coating by Chemical Grafting of Graphene  Oxide” focus on superhydrophobic surface.

The problem of dealing with icing of wires overhead lines power transmission is well known. High humidity, harsh temperature fluctuations, winds are factors due to which the intensity of icing of overhead line wires increases significantly. Due to the icing of the wires, the weight of which increases by several times, all elements of overhead lines are subjected to additional mechanical loads.

The review article covers this topic in sufficient detail, shows the  of  anti-icing mechanism and Influencing factors for this process. As a result, I will recommend the publication of this manuscript after minor revision.

Comments:

1. In the review article, little is said about the chemical structure of the substances on which icing occurs. How does the icing depend on the structure?

2. I would like to see more numerical values in the article, a comparison with industrial anti-icing coatings.

3. How does humidity affect icing, how will it change? It would be nice to add information about it.

Reviewer 3 Report

Please find my comments attached.

Reviewer 4 Report

The manuscript fits the journal very well, however it needs revision, after which it can be reconsidered for acceptance by the journal.  Below are comments and suggestions to help the authors improve it:

1\ The words in the title are not presented in a consistent way: some are spelled with low-case letters, while some others - with capital letters. Please fix

2\ The following  papers are recommended to be mentioned and cited in this review:

Progress in Chemistry 2017, 29, 102-118.    doi:  10.7536/pc161015

Cold Regions Science and Technology 2010, 62, 29–33

These are either a very useful review paper or one of papers directly related to overhead power line icing. Thus, they deserve attention.

3\ line 177, page 5: when mentioning the poor durability of superhydrophobic coatings, the authors may refer to the following papers:

Materials Characterization 2013, 82, 9-16. doi: 10.1016/j.matchar.2013.04.017

Langmuir 2011, 27, 25-29.  doi:  10.1021/la104277q

4\ Reference list: please make sure that all references in the list have volume number italicized, according to the MDPI style

5\ Reference list: please make sure that all journal titles are properly abbreviated in the list, in accordance with the MDPI style

6\ Figure 12: figure caption needs improvements: all panels must be described in detail: (a) to (j).

7\ page 12, lines 406-407: Here, the authors mention organo-silane related coatings (such as FAS or their likes) which are still widely used as hydrophobizing agents for anti-icing coatings. In this light, it is recommended that the authors discuss the following article which recently showed that such silane layers (similar to OTS, FAS and their likes) are not very stable on metal surface when they contact water for long periods of time:

Materials 15 (2022) 1804.   doi: 10.3390/ma15051804

The work was done on aluminium surfaces. And even though in the present review, the authors tend to discuss rough superhydrophobic surfaces, similar phenomena are expected on both flat and rough surfaces (degradation of OTS/FAS in contact with water). Also, it is useless to mention that long contact with water or ice is expected to lead to somewhat similar concequences for OTS/FAS layers. Which is why , mentioning this aspect (poor durability of silane layers in long-term contact with water) seems very important for the present review.

Round 2

Reviewer 4 Report

The manuscript needs revision, as the authors neglected quite many points:

1\ In refs. 136,125,111,137 (and possibly in many others?) ,chemical formulas are presented incorrectly (with no subscripts) .Please check and fix

2\ In many refs [for example, 133-134, and many others], journal titles must be properly abbreviated. Please check and fix

3\ Refs. 132 and 136: journal title must be "ACS Appl. Mater. Interfaces"

4\ in ref. 131: article number is mission. It must read:   Materials 2022, 15, 1804 (please check and fix)

5\ Refs. 127-128: journal titles must be properly abbreviated. Check and fix

Round 3

Reviewer 4 Report

The manuscript was revised and now it can be accepted